# OpenReview forum: "Revealing and Reducing Gender Biases in Vision and Language Assistants (VLAs)"
_ICLR.cc/2025/Conference — ICLR 2025 Poster_

### Official Review · Reviewer_MdLK · 2024-10-27

**Soundness:** 3
**Presentation:** 4
**Contribution:** 3
**Rating:** 8
**Confidence:** 4

**Summary:**

The paper benchmarks the performance in 22 open-source VLAs such as LLaVa, BakLLaVa, and InternVL alongside their variants on four datasets - FairFace, MIAP, Phase and PATA to identify the ingrained gender bias based on occupational, personality and work-related skills in such vision-language models. The experimental setup is Visual Question Answering(VQA) which sends as inputs an image which does not contain any specific information which would allow the model to deduce the person's occupation with the text input being an objective question requiring the model to solely deduce the occupation, personality or skill of a person based on their gender. The work concludes that a majority of these models assign more negative traits such as "stubborn", "moody" and "arrogant" to males as compared to positive traits being assigned to females. Moreover, positive skills are majorly associated with males replicating the imbalances in occupations in the real world. In this regard, the authors aim to mitigate this inherent bias through classical techniques such as - full finetuning, Low-Rank Adapter finetuning, prompt engineering and alpha-pruning. Amongst these techniques, they identify the best mitigation in full finetuning and LoRA finetuning with LoRA preserving the performance of the original model while full finetuning reducing bias to the minimum. Additionally, they observe that prompt engineering and alpha-pruning(over multiple alpha values) do not push the model towards reducing bias.

**Strengths:**

The main strengths of this paper:
1) The authors address a longstanding problem of extending the investigation of gender bias mitigation from LLMs towards VLMs. The research question has been defined structurally with the specific gender bias they investigate, i.e. its significance arising from personality, occupational and work-related traits.
2) The benchmark comprising the 22 VLAs and their performance assessments on four different datasets - MIAP, FairFace, PATA and Phase seemed particularly extensive and an elaborate amount of experimentation to me.
3) The appropriate conclusion drawing from the performance of the VLAs as well as the metric to measure the gap between the mean of answers directing to males and the mean of answers directing to females, seemed interesting. The use of prior evaluation studies to demonstrate how VLAs reflect the imbalance in current societies over gender is quite intuitive.

Other positive points include the good structure of the paper as well as relevant datasets used for the evaluation study.

**Weaknesses:**

A few weaknesses which seemed to be standing out to me:
1) While reading through the paper and some related work, I came across another paper "Gender Bias in Vision-Language Assistants" performing a similar study using the datasets used in this paper. I would suggest citing this work as well since the work seems to be aligned with the current study as well as demonstrating similar outcomes to the current paper. The authors could discuss how this work improves upon or contrasts with that work.
2) Some additional insights into the exact process of finetuning seemed to be missing from the paper such as the training set, the ground truths used for evaluation etc.
3) Another method used for bias mitigation is causal mediation analysis(Vig et al. 2020), which has been previously used in the cases of Language Models. This method can be extended to VLAs primarily to get a more concrete insight into the gender bias ingrained in these models. The authors could maybe shed some light on how that method can be used for bias mitigation.
4) In section 4.3, Qualitative results, where it is mentioned that "the distribution over the three answers becomes somewhat uniform" -  Since the answers should be skewed towards "unsure", this would demonstrate the models become "more careful" in answering such questions with confidence. Additionally, it seems to me from Figure 4, that none of the methods alleviates the model's tendency to respond with "unsure", only increasing the distribution value for "no" which could also be arising from hallucination. As mentioned by the authors, "unsure" would be an answer to be picked to avoid gender-sensitive questions, I feel debiasing should be targeted towards increasing the probability of answering "unsure" which would indicate that it needs more occupational or personality-related description to converge on a final answer related to the image.

**Questions:**

Some questions which I would like answered:

1) In section 2.3, prompt groups "We manually identify 21 suitable skill descriptions..." Is there any concrete backing to this identification process drawn from any other study?

2) Some examples can be drawn which demonstrate the different answers generated by the models to the same question asked.

3) In section 4.3, Qualitative results, where it is mentioned that the distribution over the three answers becomes somewhat uniform, can some insight be provided into how it exactly fixes the bias present in the models? Also drawing from my fourth point in weaknesses, I would like some elaboration on the implications of "no" increasing as a response to the questions asked.

---

> ### Author Response · Authors · 2024-11-21
> **Author Response to Reviewer MdLK (1/2)**
>
> We would like to thank the reviewer for the insightful assessment of our work and the helpful comments. We are happy that the reviewer recognizes the “extensive and an elaborate amount of experimentation” in our work and mentions a “good structure of the paper” as a strength. In the following, we address the concerns and questions raised by the reviewer.
>
> > While reading through the paper and some related work, I came across another paper "Gender Bias in Vision-Language Assistants" performing a similar study using the datasets used in this paper. I would suggest citing this work as well since the work seems to be aligned with the current study as well as demonstrating similar outcomes to the current paper. The authors could discuss how this work improves upon or contrasts with that work.
>
> Thank you for pointing out this related work. In contrast to our study, the mentioned work focuses on skills alone and does not include further aspects, in particular personality traits and occupations. In addition, we have taken additional measures to improve the validity of our study, most importantly removal of images containing occupation-related information. Finally, the set of models evaluated in our study is much larger in comparison (22 vs. 16 models) and considers more recent model series such as InternVL 2 and Bunny.
>
> We would like to point out that [1] is a non-archival extended abstract and has been presented at an ECCV workshop. We included this concurrent work as a reference in our related work section in the revision (l. 520), and briefly commented on the differences to our study.
>
> > Some additional insights into the exact process of finetuning seemed to be missing from the paper such as the training set, the ground truths used for evaluation etc.
>
> The training set consists of prompts constructed in the same way as described in Sec. 2.3, but only using the prompt components as well as personality traits, skills, or occupations in the training split, which is tabulated in Appendix A. For evaluation, we conduct the same analysis as in Sec. 4.2 (the method is stated in Sec. 3.1), but only using prompt components and traits/skills/occupations in the test split. Images used for training are different from those in VL-Gender (used in Sec. 4.2), and will be released together with our code and data in case of acceptance. These details are stated in Sec. 4.3 (ll. 373-404).
>
> Given precise suggestions in this matter, we are happy to revise our paper accordingly.
>
> > Another method used for bias mitigation is causal mediation analysis(Vig et al. 2020), which has been previously used in the cases of Language Models. This method can be extended to VLAs primarily to get a more concrete insight into the gender bias ingrained in these models. The authors could maybe shed some light on how that method can be used for bias mitigation.
>
> We agree that causal mediation analysis would be an interesting method to consider when debiasing models. However, we see two primary challenges when transferring the causal mediation analysis setup as described in [2] to our study:
>   1. Causal mediation analysis requires the set-gender operation, which in our case translates to a “set-personality-trait” or “set-skill” operation. For occupations, this would be straightforward to implement, but finding images which unambiguously display personality traits and skills are non-trivial to construct. In fact, we make a point that these can, in principle, not be deduced from images alone. Therefore, significant methodological innovation is required to make causal mediation analysis in our setting.
>   2. Causal mediation applied to VLAs would be significantly more expensive than applied to LLMs, because of the additional image loading and encoding step. This makes a systematic exploration which neurons or attention heads contribute most to gender bias very costly.
>
> Given these challenges, we consider applying causal mediation analysis to debiasing VLAs an interesting direction for future work.

---

> > ### Comment · Reviewer_MdLK · 2024-11-21
> >
> > Thank you for the satisfying answer to my questions, and your enthusiasm in addressing the concerns in the paper as well as elaborating on them extensively. I will be updating my score

---

> ### Author Response · Authors · 2024-11-21
> **Author Response to Reviewer MdLK (2/2)**
>
> > In section 4.3, Qualitative results, where it is mentioned that "the distribution over the three answers becomes somewhat uniform" - Since the answers should be skewed towards "unsure", this would demonstrate the models become "more careful" in answering such questions with confidence. Additionally, it seems to me from Figure 4, that none of the methods alleviates the model's tendency to respond with "unsure", only increasing the distribution value for "no" which could also be arising from hallucination. As mentioned by the authors, "unsure" would be an answer to be picked to avoid gender-sensitive questions, I feel debiasing should be targeted towards increasing the probability of answering "unsure" which would indicate that it needs more occupational or personality-related description to converge on a final answer related to the image.
>
> Please see the general response for our answer.
>
> > In section 2.3, prompt groups "We manually identify 21 suitable skill descriptions..." Is there any concrete backing to this identification process drawn from any other study?
>
> One of our aims was to create the list of skills in a data-driven way, as we are not aware of any corpus-backed skills wordlist. Therefore, we extracted n-grams containing “ability to” from a resume corpus, which is likely to include a relevant number of skills. However, one main challenge is to separate soft skills from hard skills, meaning some concrete knowledge or capability that can be learned similar to occupations. To select suitable soft skills which cannot be inferred from images alone, we had to resort to manual extraction. We are not aware of a fully automatic method to extract soft skills while filtering hard skills, and developing such a method is out of scope for this study.
>
> Many skills manually extracted by us are echoed in relevant literature on workplace gender bias, for example [3] (in study 1) evaluates “leadership ability”, “independent”, “hard working”, “works well under pressure”, “analytical”, “competent”, and “good listener”.
>
> A systematic matching of prescriptions from [3] and skills in our study is in the following table:
>
> | **Rudman et al. (2012)** | **Our study (ability to …)**          |
> |--------------------------|---------------------------------------|
> | Leadership ability       | Lead                                 |
> | Independent              | Work independently                   |
> | Hard working             | Work well, work effectively          |
> | Works well under pressure| Work under pressure                  |
> | Analytical               | Use logical approaches               |
> | Competent                | Work well                            |
> | Cooperative              | Work in teams                        |
> | Good listener            | Communicate effectively, interact professionally |
>
> Although there is a matching of some skills, our list is expanded in comparison to [3] while at the same time the list in [3] contains many attributes that we include as personality traits. This underscores the need to create a new list of skills specifically for this study.
>
>
> ## References
> [1] Girrbach et al.: *“Gender Bias in Vision-Language Assistants”*. In ECCV Workshop DarkSide of GenAIs and Beyond (2024)\
> [2] Vig et al.: Causal Mediation Analysis for Interpreting Neural NLP: The Case of Gender Bias. In arXiv, 2020\
> [3] Rudman et al.: *Status incongruity and backlash effects: Defending the gender hierarchy motivates prejudice against female leaders*. In Journal of experimental social psychology, 2012

---

### Official Review · Reviewer_Xw5N · 2024-10-30

**Soundness:** 3
**Presentation:** 3
**Contribution:** 3
**Rating:** 6
**Confidence:** 4

**Summary:**

The paper investigates gender bias in vision-language assistants (VLAs) by analyzing 22 widely used open-source models, focusing on associations with personality traits, skills, and occupations. It presents comprehensive experimental results, demonstrating the presence of biases across multiple models and offering a detailed comparison of bias patterns. The findings reveal that VLAs frequently replicate human biases, associating positive traits more often with women and negative traits with men, while also reflecting gender imbalances in occupational roles observed in real-world data. The study evaluates several debiasing techniques, concluding that fine-tuning provides the best balance between reducing bias and preserving model performance.

**Strengths:**

1. The paper presents a unique approach to analyzing gender bias in VLAs by comprehensively examining bias across 22 open-source models.

2. The study is well-executed, featuring a detailed experimental design and thorough experimental results.

3. This work holds significant value for the field of fairness in VLMs by highlighting the presence of gender bias in widely used VLAs and offering effective strategies for mitigating such biases.

**Weaknesses:**

1. In the datasets used by the authors, there are no true labels, meaning that all evaluations rely on open-ended questions. Therefore, responses of "yes" or "no" are not necessarily correct answers, and choosing "unsure" might actually be a fairer response. It would be helpful if the authors discussed how they accounted for this limitation in their analysis, such as the analyze the frequency of "unsure" responses across different demographic groups.

2. In this evaluation process, while comparing the number of "yes" responses across genders can indeed be used to evaluate bias, the open-ended nature of the questions might allow other visual factors in the images, such as attire or accessories, to influence the results. For example, a model may be more likely to associate positive personality traits with someone wearing glasses and neatly dressed, regardless of gender, which could weaken the apparent impact of gender bias. To more accurately assess the effect of gender bias, would a more focused evaluation method be possible, or could additional image datasets with true labels be supplemented to evaluate from both perspectives?

3. In the authors' mitigation experiments, the various methods tested seem to create a more balanced distribution of "yes" and "no" responses for the test images (as shown in Figure 4). However, it is possible that neither "yes" nor "no" is always the most accurate answer; in cases where the image lacks sufficient information, "unsure" might be the most appropriate response. Could the authors provide a justification for focusing on balancing "yes" and "no" responses rather than encouraging "unsure" responses? Additionally, analyzing the distribution of "unsure" responses before and after debiasing might offer further insight.

**Questions:**

1. In Section 2.1, it would be helpful to present the statistics of the evaluation set of 5,000 images, including the distribution of various labels. This additional information would clarify the dataset composition and support a more detailed understanding of the evaluation setup.

2. In Section 2.2, the authors attempt to remove biased images using InternVL2-40B and GPT-4V; however, the entire removal process relies exclusively on these pre-existing VLMs without any supervised labels or manual verification. It may be beneficial for the authors to consider additional validation methods for their image selection process, such as conducting a small-scale human evaluation on a subset of the images to verify the effectiveness of the removal.

3. In Figure 1, some images still appear to contain occupation-related information, which could introduce inaccuracies in the evaluation results and create ambiguity about whether observed unfairness is genuinely due to gender bias. Do these examples come from your cleaned dataset?

4. For prompt tuning mitigation method, what initial hard prompt was used for fine-tuning?

5. How is the significant bias score calculated, based on sample mean? What is the sample size?

6. In Figure 3, the legend currently uses shades to represent different methods, but clarity could be improved by using consistent colors for each method throughout the figure. Alternatively, if distinct colors are used, labeling each method directly on the figure may help with readability. Additionally, could the authors clarify why significant bias appears to reach zero for some models after optimization? For example, in the personality traits evaluation, LoRA tuning for llava_1.6_mistral_7b seems less effective, while full fine-tuning achieves optimal results. What factors might contribute to such a large discrepancy in optimization effects for the same model?

7. In the dataset posed by the authors, it is theoretically difficult to infer relevant information purely from images, so, in my understanding, the "unsure" option would be a more correct answer than "yes" or "no." However, in the optimization loss function, it appears that the "unsure" option is not considered.

8. Some relevant papers discussing gender bias in large vision-language models (LVLMs) appear to be missing from the Related Works section. For example:
    * Zhang, Jie, et al. "VLBiasBench: A Comprehensive Benchmark for Evaluating Bias in Large Vision-Language Models." arXiv preprint arXiv:2406.14194 (2024). This paper introduces a synthetic benchmark for systematically evaluating bias across various LVLMs (including gender bias evaluation), could provide additional context to your findings.
    * Wu, Xuyang, et al. "Evaluating Fairness in Large Vision-Language Models Across Diverse Demographic Attributes and Prompts." arXiv preprint arXiv:2406.17974 (2024). This study examines fairness across multiple demographic attributes, including gender bias in occupation prediction, using labeled data from the FACET dataset. It would be helpful if you could discuss the differences between test strategies based on labeled datasets versus open-ended question scenarios, as each approach may impact performance and insights into gender bias.


Typo:
* Finetuning => fine-tuning
* Tradeoff => trade-off
* Line 91, gender per se ?

---

> ### Author Response · Authors · 2024-11-21
> **Author Response to Reviewer Xw5N (1/3)**
>
> We would like to thank the reviewer for the thorough assessment of our work and the constructive comments. We are happy that the reviewer sees our study as “well-executed” and presenting a “a unique approach to analyzing gender bias in VLAs”, holding “significant value for the field of fairness in VLMs”. Below, we respond in detail to the concerns and questions raised by the reviewer.
>
> >   * In the datasets used by the authors, there are no true labels, meaning that all evaluations rely on open-ended questions. Therefore, responses of "yes" or "no" are not necessarily correct answers, and choosing "unsure" might actually be a fairer response. It would be helpful if the authors discussed how they accounted for this limitation in their analysis, such as the analyze the frequency of "unsure" responses across different demographic groups.
> >   * In the authors' mitigation experiments, the various methods tested seem to create a more balanced distribution of "yes" and "no" responses for the test images (as shown in Figure 4). Could the authors provide a justification for focusing on balancing "yes" and "no" responses rather than encouraging "unsure" responses? Additionally, analyzing the distribution of "unsure" responses before and after debiasing might offer further insight.
>
> The reviewer points out that “unsure” is the most suitable answer to the prompts in our study. We indeed agree to it in several places in our paper, for example ll. 139-143. However, we also show in Fig. 11 (Appendix D.3) that most models in our study do not choose “unsure” as answer. Therefore, our analysis of gender bias focuses on predicting “yes”. Concretely, we give each prompt and image in our dataset and collect the output probabilities for “yes”. Then we compare the probabilities for answering “yes” for female-gendered images to the probabilities for answering “yes” for male-gendered images to find gender bias. Finally, we would like to note that a model that predicts “unsure” in most cases will also be less biased in our analysis, because this reduces the probability mass assigned to the “yes” and “no” options.
>
> For our additional experiments regarding encouraging "unsure" instead of "yes" and "no", please see the general response.
>
> > In Section 2.1, it would be helpful to present the statistics of the evaluation set of 5,000 images, including the distribution of various labels.
>
> Please see our general response for our answer to this request.
>
> > For prompt tuning mitigation method, what initial hard prompt was used for fine-tuning?
>
> In our debiasing study, we employ two prompt-based debiasing methods, prompt tuning and prompt engineering. In prompt engineering, we add 3 prompt prefixes from [1] to prompts, but these prompts are taken as they are and not changed. For prompt tuning, we add 20 trainable embeddings after the BOS token. These embeddings are trained with the loss stated in Eq. (2). Their weights are initialized from a normal distribution with mean 0 and standard deviation 0.02.
>
> > In Figure 1, some images still appear to contain occupation-related information, which could introduce inaccuracies in the evaluation results and create ambiguity about whether observed unfairness is genuinely due to gender bias. Do these examples come from your cleaned dataset?
>
> We intended to show images from the original dataset, as in the diagram in Fig. 1, dataset debiasing is shown on the bottom. The images were intended to be inputs to the dataset debiasing. However, we acknowledge that this can be confusing to some readers, as indeed, three of the shown images are filtered by our method.
>
> Therefore, we replaced them with other images that do not show occupation-related information and are hence not filtered. The model figure is updated in the revision.
>
> > How is the significant bias score calculated, based on sample mean? What is the sample size?
>
> Our method to determine if a model exhibits significant bias tests if there is a significant difference in the distributions of P(“yes”) for female-gendered and male-gendered images. We give details on our method in Sec. 3.1 and in Eq. (1) in particular. To determine if there is a significant difference between P(“yes”) for female-gendered images and for male-gendered images, we use a two-sample t-test. The size of both samples is 2500, as our dataset (VL-Gender) contains 5000 images, half of them male-gendered and the other half female-gendered. We have made the dataset composition, including gender labels, more clear in Appendix I.

---

> ### Author Response · Authors · 2024-11-21
> **Author Response to Reviewer Xw5N (2/3)**
>
> >   * In this evaluation process, while comparing the number of "yes" responses across genders can indeed be used to evaluate bias, the open-ended nature of the questions might allow other visual factors in the images, such as attire or accessories, to influence the results. For example, a model may be more likely to associate positive personality traits with someone wearing glasses and neatly dressed, regardless of gender, which could weaken the apparent impact of gender bias. To more accurately assess the effect of gender bias, would a more focused evaluation method be possible, or could additional image datasets with true labels be supplemented to evaluate from both perspectives?
> >   * In Section 2.2, the authors attempt to remove biased images using InternVL2-40B and GPT-4V; however, the entire removal process relies exclusively on these pre-existing VLMs without any supervised labels or manual verification. It may be beneficial for the authors to consider additional validation methods for their image selection process, such as conducting a small-scale human evaluation on a subset of the images to verify the effectiveness of the removal.
>
> Dataset bias is a serious problem when analyzing bias in general, as observed by the reviewer. We address this issue explicitly in Sec. 2.2. As measures to reduce dataset bias, we include images from various source datasets in our study and automatically remove images showing occupation-related content. In this way, we aim to include images of sufficient diversity and also remove the principal confounding axis, namely occupation.
>
> Clearly, it is possible to remove more images based on other attributes. However, this would require a way to determine what confounders models are actually susceptible to. Furthermore, such confounders are likely model-specific. As we are currently not aware of a practical method to determine such confounders, we leave a more targeted evaluation that takes additional confounders beyond occupation into account for future work.
>
> As mentioned, we removed images showing occupation-related information in an automatic way. To confirm the validity of this method, three volunteers (not authors of this study, 2 of them male and one of them female) independently labeled a subset of 100 images. These 100 images were selected in a stratified fashion, the subset contains 50 images where the mentioned probability of containing occupation information is greater or equal than 0.25, i.e. images that were removed, and 50 images where the probability is lower, i.e. images that are kept. Both subsets of 50 images contain each 5 images per combination of dataset and gender (5 images x 2 genders x 5 datasets).
>
> We are mainly interested in the inter-annotator agreement among human annotators, and the recall of automatic removal. For our purposes, it is only relevant that as many images as possible showing occupation-related content are removed.
>
> As an inter-annotator agreement statistic, we report Krippendorff’s $\alpha$, which amounts to 0.638, indicating substantial agreement.
> To calculate precision, recall and f1 scores, we assign the majority label among the three annotators to each image and calculate the scores in the following table:
>
> | Category | Precision | Recall | F1-Score | Support |
> |--------------------------|-----------|--------|----------|---------|
> | Doesn’t show occupation | 0.96      | 0.69   | 0.80     | 70      |
> | Does show occupation | 0.56      | **0.93**   | 0.70     | 30      |
> | Accuracy | | | 0.76 | 100     |
> | Macro Avg | 0.76 | 0.81   | 0.75     | 100     |
> | Weighted Avg | 0.84      | 0.76   | 0.77     | 100     |
>
>
> We highlight the recall of images that do not show occupation, which is 0.93. This is a very high score, showing that almost all images annotated as showing occupation (28 of 30) are also removed by InternVL2 40B. As expected, InternVL 40B also removes many images not annotated as showing occupation, hence the relatively low precision (0.56) on this class.
>
> > In Figure 3, the legend currently uses shades to represent different methods, but clarity could be improved by using consistent colors for each method throughout the figure. Alternatively, if distinct colors are used, labeling each method directly on the figure may help with readability
>
> Using the same colors for methods instead of shades is certainly a sensible alternative to the current design. Our intention is to provide a consistent coloring of model series throughout the whole paper. This is especially relevant given that the supplementary contains many more figures than the main paper.

---

> ### Author Response · Authors · 2024-11-21
> **Author Response to Reviewer Xw5N (3/3)**
>
> > Additionally, could the authors clarify why significant bias appears to reach zero for some models after optimization? For example, in the personality traits evaluation, LoRA tuning for llava_1.6_mistral_7b seems less effective, while full fine-tuning achieves optimal results. What factors might contribute to such a large discrepancy in optimization effects for the same model?
>
> Our hypothesis is, that, most likely, the success of a debiasing method depends on how bias is encoded in the model. The more bias is encoded in a distributed manner in various model components, the harder it may be to successfully mitigate the bias by relatively low capacity methods such as LoRA. Instead, in such cases full finetuning may be the only effective method to tackle gender bias. Another consideration is how well the debiasing generalizes to different prompt variations, which we explicitly target in our study by separating all prompt components into non-overlapping train and test splits. When LoRA is not as effective as full-finetuning, it may have picked up on patterns in the training prompts, which are not present in the test prompts.
>
> > Some relevant papers discussing gender bias in large vision-language models (LVLMs) appear to be missing from the Related Works section. For example:
> >   * Zhang, Jie, et al. "VLBiasBench: A Comprehensive Benchmark for Evaluating Bias in Large Vision-Language Models." arXiv preprint arXiv:2406.14194 (2024). This paper introduces a synthetic benchmark for systematically evaluating bias across various LVLMs (including gender bias evaluation), could provide additional context to your findings.
> >   * Wu, Xuyang, et al. "Evaluating Fairness in Large Vision-Language Models Across Diverse Demographic Attributes and Prompts." arXiv preprint arXiv:2406.17974 (2024). This study examines fairness across multiple demographic attributes, including gender bias in occupation prediction, using labeled data from the FACET dataset. It would be helpful if you could discuss the differences between test strategies based on labeled datasets versus open-ended question scenarios, as each approach may impact performance and insights into gender bias.
>
> Both works discuss social bias from different perspectives. In [2], Wu et al. evaluate gender bias as differences in prediction recall on occupation prediction. For example, if a model performs worse on predicting the class “gardener” for female-labeled images than for male-labeled images, we may conclude that the model associates the occupation “gardener” more with males than with females. In contrast to [2], our method takes a more direct approach, directly querying VLAs for their association of persons with personality traits, skills and occupations. Another important difference to our work is that we eliminate occupation-related information from our dataset, while occupation-related information is intentionally included in FACET. Thus, our method enables an analysis that detects gender bias beyond occupation bias, while also not conflating performance of models in detecting occupations with gender bias.
>
> The main difference between our work and [3] is that they use synthetic images instead of natural images, and they evaluate sentiment in free-text generation and accuracy in close-form evaluation. Note, that in contrast to [2], Zhang et al. don’t look at accuracy differences between genders, but on the accuracy on prompts directly designed to target gender bias. While both approaches have their respective advantages and limitations, our study stands out in its detailed analysis of gender bias in precisely specified personality traits, skills and occupation.
>
> In the revision, we added [2] as an additional reference in l. 498, but [3] was already cited in l. 503 of the original version. Both studies are also included in the systematic comparison in Appendix N, which is new in the revision.
>
> > Typo:
> >  * Finetuning => fine-tuning
> >  * Tradeoff => trade-off
> >  * Line 91, gender per se ?
>
> According to the OED, “fine-tuning” and “trade-off” are indeed the correct spellings of the mentioned words. We have adjusted the spelling in the revision.
>
> Furthermore, we have removed “per se” in l. 91, as we think its addition does not add relevant information, and, as the reviewer noted, may be confusing to readers.
>
>
> ## References
> [1] Howard et al.: Uncovering bias in large vision-language models at scale with counterfactuals. In arXiv, 2024\
> [2] Wu et al.: Evaluating Fairness in Large Vision-Language Models Across Diverse Demographic Attributes and Prompts. In arXiv, 2024\
> [3] Zhang et al.: VLBiasBench: A Comprehensive Benchmark for Evaluating Bias in Large Vision-Language Models. In arXiv, 2024

---

> > ### Comment · Reviewer_Xw5N · 2024-11-21
> >
> > Thank you very much for the authors' response and the new results. Most of my concerns have already been addressed. I hope the authors can incorporate these improvements to further enhance the paper. I will update my score accordingly.

---

### Official Review · Reviewer_vjUX · 2024-11-03

**Soundness:** 4
**Presentation:** 3
**Contribution:** 3
**Rating:** 6
**Confidence:** 4

**Summary:**

This paper investigates gender bias in 22 open-source instruction-tuned VLMs (a.k.a. VLAs) across three different categories: personality traits, skills, and occupations. The authors curate a dataset for these analyses, and show that VLAs follow real-world gender imbalances for some categories (e.g. occupations) but less for other categories (e.g. personality traits). The authors then investigate 5 techniques to reduce gender bias in 5 models, and find full fine-tuning to return the better trade-off between overall performance and gender bias.

**Strengths:**

1. The paper combines and extend different datasets for gender (and ethnicity) bias.
2. Several (22) open-source VLAs are evaluated, showcasing overall similar patterns.
3. Five different techniques are tested to reduce gender bias, with results on the trade-offs of each of them w.r.t. gender bias and overall performance.

**Weaknesses:**

1. One main weakness of the proposed approach — as pointed out by the authors — is the inability to test API-based VLAs
2. Another question I had while reading the paper was how this study differentiates from previous ones. This was only partially answered towards the end of the paper (Related Work Section), but it would be very helpful to have a more systematic comparison with previous work and how this study clearly differs from it (e.g. by creating a table summarizing key differences), as the results echoes findings that had previously been published.
3. Among them, a comparison and discussion that is missing is around the work of Cabello et al. (EMNLP 2023), which shows that fine-tuning VLMs on gender-neutral data led to similar or better performance whilst reducing bias. Given that you find fine-tuning to be the better method, such comparison (or at least a discussion) would be very relevant to include (e.g. can the two approaches be combined?)

---
Cabello et al. Evaluating Bias and Fairness in Gender-Neutral Pretrained Vision-and-Language Models. EMNLP 2023.

**Questions:**

1. Have you tried assessing your 22 models using approaches that could be applied to API-only models? It would at least be informative to know how your conclusions would change in that case.
2. In L313-314, you claim that “positive traits, however, are not attributed to one single gender.” On the other hand, Fig 2 (top) shows that — besides “humble” — all the other traits are mostly associated with female gender. Can you provide a more detailed explanation of how you interpret these results to support the current statement?
3. In Section 4.3, you split the data into training and test. How do you perform hyper-parameter tuning without a validation set?

---

> ### Author Response · Authors · 2024-11-21
> **Author Response to Reviewer vjUX (1/2)**
>
> We would like to thank the reviewer for the positive and constructive feedback, highlighting our contributions in combining and extending different datasets, evaluating several VLAs, and testing five different techniques to reduce gender bias. In the following, we respond to all questions asked by the reviewer.
>
> >   * One main weakness of the proposed approach — as pointed out by the authors — is the inability to test API-based VLAs
> >   * Have you tried assessing your 22 models using approaches that could be applied to API-only models? It would at least be informative to know how your conclusions would change in that case.
>
> Our method relies on logits to calculate if there is a significant gender bias (Sec 3.1). Most importantly, we test if there is a significant difference in the distributions of probabilities for answering “yes” to given prompts between male-gendered and female-gendered images. Therefore, access to the logits is necessary to assess gender bias in the proposed way. Unfortunately, currently API models generally do not give access to logits.
>
> One possible alternative is to calculate the ratios of actually predicted answers (“yes”, “no”, or “unsure”) for male-gendered and female-gendered images. Then, the difference between ratios for predicting “yes” to our prompts for male-gendered and female-gendered images is an alternative way of assessing gender bias, which only relies on concrete answers and not on probabilities. In the new Appendix L, we present results using this method in an analogous fashion to Fig. 2. We show that our conclusions also remain unchanged with this method, but applying it to API models is infeasible due to cost constraints.
>
> However, a main advantage of our method in comparison to directly evaluating the frequency of given answers is that we can detect differences even when the generated answer is always the same. Biases may in this case not surface, but they can still be reflected in more or less certain answers. We think this is an important property of our method, because we aim at quantifying gender bias in a generalizable way, and not only at quantifying gender bias with respect to the prompts that are evaluated in our study. To this end, we also need to be able to detect latent associations that may only surface in different settings.
>
> > Another question I had while reading the paper was how this study differentiates from previous ones. This was only partially answered towards the end of the paper (Related Work Section), but it would be very helpful to have a more systematic comparison with previous work and how this study clearly differs from it (e.g. by creating a table summarizing key differences), as the results echoes findings that had previously been published.
>
> We summarized the the most relevant related work in a systematic and structured way in the new Appendix N, see in particular Table 12. We focus on which models are evaluated, which images and demographics are included in the study, and the tasks used to assess bias.
>
> Our study stands out in comparison to previous work in its unprecedented scale (evaluating 22 open-source models), comprehensiveness of evaluation (using natural images from 4 different datasets), and evaluation of specific bias concepts. The last aspect contrasts with previous work that predicts image content such as occupation or gender, which we show that models are generally good at. This table and summary are included in the revision in the new Appendix L, discussing additional related work.
>
> > Among them, a comparison and discussion that is missing is around the work of Cabello et al. (EMNLP 2023), which shows that fine-tuning VLMs on gender-neutral data led to similar or better performance whilst reducing bias. Given that you find fine-tuning to be the better method, such comparison (or at least a discussion) would be very relevant to include (e.g. can the two approaches be combined?)
>
> The work of Cabello et al. indeed presents an interesting approach to debiasing vision-language models, and in the revision we included it as reference in line 494. The main idea is to continue pretraining models on gender-neutral data, i.e. by replacing gendered words such as “woman” or “man” with gender-neutral alternatives, such as “person”. We think that this approach can be combined with the debiasing methods evaluated in our work, for example in a multi-task fashion. However, one significant challenge is to transfer the rule-based word-replacement method of Cabello et al. (which is only applied to captioning data) to instruction-tuning data required to train VLAs. Therefore, we think that combining the methods in (Cabello et al., 2023) and our methods would be an interesting future work.

---

> ### Author Response · Authors · 2024-11-21
> **Author Response to Reviewer vjUX (2/2)**
>
> > In L313-314, you claim that “positive traits, however, are not attributed to one single gender.” On the other hand, Fig 2 (top) shows that — besides “humble” — all the other traits are mostly associated with female gender. Can you provide a more detailed explanation of how you interpret these results to support the current statement?
>
> Thank you for pointing out that this formulation deserves to be improved. Due to space constraints, we do not show results for all evaluated personality traits in Fig. 2. Instead, there we only show the top 5 female-biased and the top-5 male-biased personality traits. The full set of personality traits and the respective gender biases of VLAs is in Fig. 12 in the supplementary. There, it is also visible that in addition to “humble”, “wise” and “loyal”, which are positive adjectives, are more attributed to males than to females as well. This is how we derive the conclusion quoted by the reviewer (“positive traits, however, are not attributed to one single gender.”). In line 317 in the revision, we have adjusted the wording to make this clearer, and we explicitly mention which other positive adjectives are also more attributed to males.
>
> > In Section 4.3, you split the data into training and test. How do you perform hyper-parameter tuning without a validation set?
>
> Given the computational and technical effort that is required to tune and evaluate several LLMs with billions of parameters across various scenarios, in this work we refrained from systematic hyperparameter tuning and instead manually choose hyperparameters that lead to successful optimization on the training set. We did not select or tune hyperparameters on the test set.

---

> > ### Author Response · Authors · 2024-11-30
> > **Author Response to Reviewer vjUX**
> >
> > Dear Reviewer vjUX,
> >
> > We hope our response has addressed your concerns about our paper. If there are any remaining limitations that might influence your rating, we would be happy to discuss them further with you. Thank you again very much for your time and for your valuable feedback.

---

### Official Review · Reviewer_P2y7 · 2024-11-10

**Soundness:** 4
**Presentation:** 4
**Contribution:** 3
**Rating:** 8
**Confidence:** 4

**Summary:**

This paper presents a method to estimate gender bias (wrt personality traits, skills and occupation) in vision-language assistants. The method prompts VLAs to say whether a person (cropped to just show the face) has a personality trait/skill/occupation, and looks at how the P(yes) varies between male and female groups. The authors evaluate how five debiasing methods help in mitigating gender bias in VLAs.

**Strengths:**

- **Sound methodology:** The methodology for constructing the dataset seems very thorough and well thought-out: sampling images from multiple datasets and balancing gender+ethnicity (although it would be useful to see which ethnicities were represented and the distribution), thorough prompt variation with accounting for prompt wording and choice ordering.

- **Thorough methodology and analysis:** The evaluation on 22 different models seems very thorough. Good citations for where the stereotypes corresponding to the occupation/skill biases come from. Something similar can be done for the personality traits by looking at valence scores (e.g. from the VAD Lexicon).

**Weaknesses:**

- **Insufficient literature review:** Missing citations from “gender bias in vision-language” literature. All but one paper in their review is from 2024, with no reference to previous works and findings. I would point to the following papers:
    1.  Survey of Social Bias in Vision-Language Models: https://arxiv.org/abs/2309.14381
    2. Worst of Both Worlds: Biases Compound in Pre-trained Vision-and-Language Models: https://arxiv.org/abs/2104.08666
    3. Counterfactually Measuring and Eliminating Social Bias in Vision-Language Pre-training Models: https://dl.acm.org/doi/abs/10.1145/3503161.3548396

**Questions:**

- L353: I dont think biasedness is a word, it should be “bias”
- An interesting analysis would be to see how consistent the biases of different models in the same model series are.
- Why do you refer to the vision-language models as “assistants” throughout the paper? “Vision-language models” is a well-established term for these models. Assistants seems to indicate that there is a larger system that the model is a part of (which is not the case, as far as I can tell).

---

> ### Author Response · Authors · 2024-11-21
> **Author Response to Reviewer P2y7 (1/2)**
>
> We would like to thank the reviewer for the favorable review of our work, characterizing our analysis as “sound” and “thorough”. Below, we respond to all suggestions and concerns mentioned by the reviewer and state what changes we made in the revision to address them.
>
> > Something similar can be done for the personality traits by looking at valence scores (e.g. from the VAD Lexicon).
>
> Enriching the personality traits in our study with their respective valence scores is a very interesting idea. Our study categorizes all personality traits into positive and negative adjectives, and we find that VLAs associate positive personality traits more with females, and negative personality traits are predominantly associated with males. Adding valence scores potentially provides a more nuanced perspective. Therefore, in a new Appendix J in the revision, we state valence scores from [1] for all personality traits in our study.
>
> However, we do not observe any clear patterns related to valence scores. Negative adjectives with relatively higher valence scores are “lazy” and “moody”, which are more associated with males by VLAs, but we do not see that the association between these two adjectives and males is weaker than the association between males and other male-biased personality traits. Instead, “moody” is one of the adjectives strongly associated with males. Similarly, “wise” and “enthusiastic” have the lowest valence scores among the positive personality traits, but one of them (“wise”) is more associated with males by VLAs and the other (“enthusiastic”) is strongly associated with females.
>
> Therefore, we conclude that while the models in our study show an interesting pattern regarding the coarse classification of personality traits into positive and negative, they do not replicate more nuanced patterns found when ordering personality traits on a real-valued scale.
>
> > Insufficient literature review: Missing citations from “gender bias in vision-language” literature. All but one paper in their review is from 2024, with no reference to previous works and findings. I would point to the following papers:
> >   * Survey of Social Bias in Vision-Language Models: https://arxiv.org/abs/2309.14381
> >   * Worst of Both Worlds: Biases Compound in Pre-trained Vision-and-Language Models: https://arxiv.org/abs/2104.08666
> >   * Counterfactually Measuring and Eliminating Social Bias in Vision-Language Pre-training Models: https://dl.acm.org/doi/abs/10.1145/3503161.3548396
>
> Indeed we focus our literature review on works assessing bias in very recent models, which we consider most relevant for this work. Due to space constraints, we only included a limited discussion of CLIP-type vision-language models and omitted a discussion of pre-ChatGPT generative vision-language models such as BLIP [2], ALBEF [3], or LXMERT [4]. For bias in LLMs alone, without visual capability, we refer to an extensive and recent study by Gallegos et al. [5]. We acknowledge that bias in vision-language models is an extensive field, and the references provided by the reviewers were missing. Therefore, in the revision, we also include references to the suggested papers in l. 492. We thank the reviewer for bringing these to our attention.
>
> > L353: I dont think biasedness is a word, it should be “bias”
>
> We mistakenly used “biasedness” in two places in the paper, namely in line 353 and also in line 505. In both places, we changed the word to “bias”.
>
> > An interesting analysis would be to see how consistent the biases of different models in the same model series are.
>
> We calculated the Pearson correlation of gender bias (i.e. the differences $\mu_{\text{male}} - \mu_{\text{female}}$) within each model series in our study across all personality traits, skills, and occupations. The results are shown in the new Appendix K (Fig. 21).
>
> Overall, we observe strong correlation between gender bias of models within the same series, especially for personality traits and occupations. Among models, there are two exceptions (InternVL2-1B and MobileVLM-1.7B) with different gender bias than other models in the respective series. These are the two smallest models in this study, so we assume the reason is their relatively inferior capacity. Also noticeably, correlation among models is generally lower for skills. While correlations are still positive, models within the same series are often only weakly correlated. This is especially apparent for LLaVA-1.6, LLaVA-1.5, and MobileVLM. Therefore, we conclude that gender bias in models is also task-specific, i.e. there is no single dimension of gender bias, but models can be biased in varying and not necessarily consistent ways.

---

> > ### Comment · Reviewer_P2y7 · 2024-11-21
> > **Response to Rebuttal**
> >
> > Thank you for your response. i am keeping my score the same.

---

> ### Author Response · Authors · 2024-11-21
> **Author Response to Reviewer P2y7 (2/2)**
>
> > Why do you refer to the vision-language models as “assistants” throughout the paper? “Vision-language models” is a well-established term for these models. Assistants seems to indicate that there is a larger system that the model is a part of (which is not the case, as far as I can tell).
>
> We agree that the phrase “vision-language assistant” is non-standard. Our intention is to clearly, already in the title, set our work apart from studies on bias in other vision-language models. Here, by other vision-language models, on the one hand-we mean CLIP-type models, which are not generative. On the other hand, we also mean non instruction-tuned models such as BLIP [2], Fuyu [6], or Flamingo [7]. Non-generative or non instruction-tuned models cannot be analyzed with the methods in this study, as we require models to follow prompts and give answers to arbitrary queries. Furthermore, we consider the analysis of instruction-tuned models as particularly important, as users will directly interact with these and in this way bias in instruction-tuned models is likely to have the most relevant societal implications.
>
> ## References
> [1] Mohammad: *Obtaining Reliable Human Ratings of Valence, Arousal, and Dominance for 20,000 English Words*. In ACL, 2018\
> [2] Li et al.: *Blip: Bootstrapping language-image pretraining for unified vision-language understanding and generation*. In ICML, 2022\
> [3] Li et al.: *Align before Fuse: Vision and Language Representation Learning with Momentum Distillation*. In NeurIPS, 2021\
> [4] Tan and Bansal: *LXMERT: Learning Cross-Modality Encoder Representations from Transformers*. In EMNLP, 2019\
> [5] Gallegos et al.: *Bias and fairness in large language models: A survey*. In Computational Linguistics, 2024.\
> [6] Bavishi et al.: *Fuyu-8B: A Multimodal Architecture for AI Agents*. ADEPT, 2023\
> [7] Alayrac et al.: *Flamingo: a visual language model for few-shot learning*. In NeurIPS, 2022

---

### Author Response · Authors · 2024-11-21
**Global Author Response**

We thank the reviewers for their detailed comments and constructive feedback with their favorable ratings, concretely 8 (Reviewer P2y7), 6 (Reviewer vjUX), 5 (Reviewer Xw5N), and 6 (Reviewer MdLK).

As summarized by the reviewers, we investigate gender bias in vision-language assistants and furthermore compare different techniques to debias models. We present a “unique approach to analyzing gender bias” (Reviewer Xw5N) which, as we also think, “holds significant value for the field of fairness in VLMs” (also Reviewer Xw5N). From our analysis, we draw “interesting” and “appropriate conclusions” (Reviewer MdLK) regarding “specific gender bias” (Reviewer MdLK) in personality traits, skills and occupations.

All reviewers agree on the comprehensiveness and extent of our analysis, as we include 22 open-source VLAs in our study and curate our dataset from 5 source datasets. Reviewer MdLK highlights the “particularly extensive and elaborate amount of experimentation”. Another strength of our work, according to reviewers, is its “sound methodology”, “thorough methodology and analysis” (Reviewer P2y7) and “detailed experimental design and thorough experimental results” (Reviewer Xw5N). In individual and detailed responses to each reviewer, we hope to address all remaining questions regarding our paper.

Commonly mentioned limitations of the current submission are
  1. Missing details on the dataset (Reviewer P2y7 and Reviewer Xw5N)
  2. Missing related work (Reviewer P2y7, Reviewer vjUX, and Reviewer Xw5N)
  3. Missing additional experiments for debiasing with a different target (Reviewer Xw5N and Reviewer MdLK)

We address the commonly mentioned limitations (1) and (3) here and respond to individual questions as well as to pointers to additional literature in direct answers to the reviews.

Throughout the discussion period, we will answer questions or concerns brought to our attention by the reviewers, and we are happy to further improve our paper.

---

> ### Author Response · Authors · 2024-11-21
> **Global Author Response**
>
> ## Dataset details
> We added gender and ethnicity statistics as a new Appendix I in the revision. The ethnicity distributions are in Fig. 20. Also, we make clear that our data is perfectly balanced for gender by mentioning that our data contains  2500 male-gendered and 2500 female-gendered images. Ethnicities are also balanced, although not perfectly, due to the removal of images with occupation-related content. For MIAP, ethnicity labels are not available at all. Furthermore, there are two ethnicities in the Phase dataset with an insufficient number of images after filtering images with occupation-related content, namely Middle Eastern and Southeast Asian, making the overall distribution for the Phase dataset imbalanced.
>
> ## “Unsure as debiasing target”
> Our proposed debiasing experiment was motivated by the finding that models in most cases prefer to answer “yes” or “no” instead of “unsure” (as mentioned in l. 186), and also to directly target the quantity that we use to assess gender bias, i.e. the probability of predicting “yes”.
>
> However, we acknowledge that encouraging “unsure” also deserves to be evaluated. Therefore, we repeated the debiasing experiments for the Prompt Tuning, Full Fine-tuning and LoRA Fine-Tuning methods using the following loss instead of Eq. (2): $L = 1 - p(\text{"unsure"})$.
>
> This loss maximizes the probability of predicting “unsure” in all prompts. Other aspects of the setup, i.e. splits in training and test attributes and prompt components, remain unaffected. For this variation, we get the following results:
>
> | Method           | Sentiment | Skills | Occupations | MMBench    | MME               | MMMU               | SEEDBench          |
> |-------------------|-----------|--------|-------------|------------|-------------------|--------------------|--------------------|
> | Original          | 0.86      | 0.48   | 0.56        | 0.60       | 1773.55           | 0.26               | 0.50               |
> | Prompt Tuning     | 0.50      | 0.22   | 0.23        | 0.54 (-8.88%) | 1696.66 (-4.34%)  | 0.23 (-9.90%)      | 0.47 (-5.31%)      |
> | Tuning (Full)     | 0.28      | 0.20   | 0.23        | 0.56 (-6.72%) | 1675.68 (-5.52%)  | 10.26 (+1.04%)     | 0.47 (-6.66%)      |
> | Tuning (LoRA)     | 0.42      | 0.22   | 0.33        | 0.56 (-5.88%) | 1703.44 (-3.95%)  | 0.23 (-9.90%)      | 0.48 (-3.69%)      |
>
>
> In contrast to our original experiment, debiasing results are improved on all prompt groups, especially personality traits. At the same time, we see less performance degradation on general benchmarks. We hypothesize that “unsure” does not appear often as an option in the general benchmark prompts, and therefore tuning models to predict “unsure” on our prompts does not interfere much with these tasks. Another observation is that prompt tuning performs worse on debiasing in this setting, confirming our initial conclusion that Full Fine-Tuning and LoRA Fine-Tuning offer the best trade-off between debiasing performance and performance degradation on general benchmarks. As before, LoRA performs worse on debiasing, but also reduces performance on general benchmarks less.

---

### Meta-Review · Area_Chair_enqy · 2024-12-17

**Metareview:**

**Summary**
This paper systematically examines gender bias in 22 open-source vision-language models (VLAs) across personality traits, skills, and occupations. The authors find that these models often reproduce real-world gender imbalances—for example, assigning more positive traits to women and more negative traits to men, as well as reflecting skewed occupational distributions. They explore various debiasing techniques and conclude that full fine-tuning offers the best trade-off between bias mitigation and overall model performance.

**Strength**
1. The paper presents a unique approach to analyzing gender bias.
2. The study is well-executed with a strong experimental design.
3. The paper draws appropriate conclusions based on performance.

**Weakness**

1. Lack of True Labels: Open-ended questions without ground-truth labels make responses ambiguous, requiring analysis of "unsure" answers.
2. Influence of Visual Factors: Visual cues like attire or accessories may confound results, requiring more controlled evaluations.
3. Insufficient Literature Comparison: The paper lacks key citations and systematic comparisons to prior work.

**Decision**
This work holds significant value for the field of fairness in VLMs with well-executed study and a unique approach to analyzing gender bias.  Based on recommendations from reviewers (8:P2y7, 6:vjUX, 6:Xw5N, 8:MdLK), I recommend accepting this paper.

**Additional Comments On Reviewer Discussion:**

During the rebuttal period, the author resolved most of the reviewer's concerns and addressed the ambiguity with a clear reference.

Ambiguity in Evaluation: Reviewers raised concerns about the lack of ground-truth labels and "unsure" responses. The authors acknowledged the limitation and provided additional analysis and clarifications.

Visual Confounders: Reviewers noted that visual cues like attire might influence results. The authors detailed their dataset cleaning process but acknowledged remaining challenges.

Missing Prior Work: The authors added key missing citations and clarified their contributions relative to prior studies, addressing reviewer concerns.

---

### Decision · Program_Chairs · 2025-01-22

Accept (Poster)